# Cross-Talk Between Neutrophils and Macrophages Post-Myocardial Infarction: From Inflammatory Drivers to Therapeutic Targets

**DOI:** 10.3390/ijms262110575

**Published:** 2025-10-30

**Authors:** Letitia Ciortan, Razvan Daniel Macarie, Elena Barbu, Miruna Larisa Naie, Andreea Cristina Mihaila, Mihaela Serbanescu, Elena Butoi

**Affiliations:** 1Inflammation Department, Institute of Cellular Biology and Pathology “Nicolae Simionescu”, 050568 Bucharest, Romania; letitia.ciortan@icbp.ro (L.C.); miruna.naie@icbp.ro (M.L.N.); andreea.mihaila@icbp.ro (A.C.M.); mihaela.vadana@icbp.ro (M.S.); 2Department of Cardiology, Elias Emergency University Hospital, Carol Davila University of Medicine and Pharmacy, 050474 Bucharest, Romania; elena.barbu@umfcd.ro

**Keywords:** neutrophils, macrophages, cross-talk, inflammation, myocardial infarction

## Abstract

Acute myocardial infarction (MI) is a major cardiovascular event and a leading cause of mortality worldwide. Beyond the initial ischemic injury, the inflammatory and immune responses play pivotal roles in both tissue damage and subsequent healing. While the anti-inflammatory strategies targeting neutrophil-driven injury have demonstrated potential in limiting early cardiac damage, growing evidence highlights the critical role of innate immune cells beyond the acute phase. Neutrophils, traditionally associated with tissue injury, also contribute to the resolution of inflammation and initiate key repair processes. Monocytes and macrophages follow a dynamic trajectory, transitioning from pro-inflammatory to reparative states, and play essential roles in debris clearance, angiogenesis, and scar formation. In the early inflammatory phase of acute MI, immune cells such as neutrophils and monocytes are rapidly recruited and activated. While they initially amplify inflammation through the release of pro-inflammatory mediators, their subsequent transition toward anti-inflammatory and reparative phenotypes helps limit tissue damage by clearing necrotic debris from the infarcted area and contributes to the resolution of inflammation. Accumulating evidence reveals a complex crosstalk between neutrophils and macrophages post-MI, with resident macrophages being involved in neutrophil recruitment, and neutrophil-derived signals participating in monocyte recruitment and macrophage polarization, thereby coordinating the spatial and temporal phases of cardiac repair. Understanding how neutrophil-derived mediators influence macrophage responses and whether macrophage-secreted factors reciprocally modulate neutrophil behavior opens promising pathways for developing targeted therapies to limit adverse remodeling following MI. Therefore, this review aims to (i) provide an overview of the roles of neutrophils and monocytes/macrophages in the pathophysiology of myocardial infarction, (ii) explore the mechanisms of communication, particularly via neutrophil-derived secreted factors, that influence monocyte/macrophage function and impact post-MI inflammation, repair, and remodeling, and (iii) highlight the potential therapies interfering with inflammation and neutrophil/macrophage cross-talk.

## 1. Introduction

Myocardial infarction (MI), commonly referred to as a “heart attack”, remains one of the leading causes of mortality in developed countries and is an increasingly significant health burden in developing regions, second only to infectious diseases [1]. Pathologically, MI is defined as myocardial cell death resulting from prolonged ischemia, caused by an imbalance between oxygen supply and myocardial demand. In most cases, this imbalance arises from occlusive coronary atherosclerosis complicated by superimposed luminal thrombosis [2]. Rupture or erosion of an unstable atherosclerotic plaque exposes subendothelial matrix and lipid-rich contents, triggering platelet activation and thrombus formation. Although no pathogens are involved, this event elicits a strong inflammatory response driven by danger-associated molecular patterns (DAMPs) released from activated or dying vascular cells. These DAMPs engage pattern recognition receptors (PRRs) such as Toll-like receptors (TLRs) and the NLRP3 inflammasome, amplifying the recruitment and activation of neutrophils, monocytes, and lymphocytes. The occlusion of the artery due to plaque rupture leads to a lack of oxygen in the myocardium, which results in sarcolemmal disruption and myofibril relaxation [3]. In the ischemic myocardium, sterile inflammation is initiated within minutes after injury, and is essential for the clearance of dead cells and matrix debris. However, excessive or unresolved inflammation exacerbates tissue damage, expands infarct size, and contributes to adverse ventricular remodeling, highlighting the dual nature of inflammation as both reparative and detrimental in myocardial injury.

Depending on the territory of the left ventricle (LV) affected by infarction, the cardiac function may be compromised. Because the myocardium only has minimal regenerative capacity, the infarcted area heals through scar formation, often leading to adverse remodeling, characterized by chamber dilation, compensatory hypertrophy of the surviving myocardium, and progressive cardiac dysfunction [3].

Scar formation is a stepwise process that begins immediately after ischemic cell death, and is initiated by the release of DAMPs from necrotic cardiomyocytes and cardiac fibroblasts that activate innate immune pathways, initiating an acute sterile inflammatory phase. Together with a small number of monocytes, dendritic cells, mast cells, and regulatory T/B cells, macrophages orchestrate the initial inflammatory response following tissue injury or infection. Intracellular signaling and cross-talk between C–C chemokine receptor 2 (CCR2)^−^ and CCR2^+^ cardiac resident macrophages is critical in ensuring the propagation of the inflammatory response and ensuring leukocyte recruitment into the affected tissue [4,5]. Among the immune cells recruited during sterile inflammation, neutrophils and macrophages are the predominant effectors shaping the outcome of myocardial injury. Within the first 24 h, and up to day 3 post-injury, myeloid cells contribute to tissue inflammation, while lymphocytes (day 3–day 7) are recruited to exert immune regression and anti-inflammatory reparative processes, including wound healing and scar formation [6].

Neutrophils are the earliest responders, infiltrating the ischemic myocardium within hours after infarction. They secrete cytokines and chemokines that orchestrate the recruitment of circulating monocytes. After infiltrating the infarct zone, monocytes initially amplify the inflammatory response while also promoting tissue clearance by phagocytosing dead cells and facilitating the transition toward resolution. These monocytes differentiate into macrophages that display remarkable plasticity: pro-inflammatory macrophages (classically referred to as M1-like) amplify injury by sustaining cytokine release and phagocytosing dead cells, while reparative (M2-like) macrophages promote resolution of inflammation, angiogenesis, and scar formation. The tightly regulated temporal sequence of neutrophil clearance followed by macrophage-driven repair is essential for effective healing. Equally important, the dynamic balance between pro-inflammatory and reparative macrophage subsets determines whether post-infarction remodeling proceeds adaptively or progresses toward adverse ventricular dysfunction and heart failure. Among the key factors influencing macrophage polarization and function is the direct and indirect cross-talk with neutrophils, mediated by paracrine signals and released effector molecules. This review, therefore, aims to highlight the mechanisms, timing, and outcomes of neutrophil–macrophage interactions in the context of acute myocardial infarction.

## 2. Role and Diversity of Neutrophils in Myocardial Infarction

Neutrophils are the most abundant circulating leukocytes in humans, produced continuously in bone marrow (BM) by granulopoiesis, acting as the first responders to infection and sterile inflammation. During infection or injury, the BM produces more neutrophils through emergency or reactive granulopoiesis to meet the host’s high demand. The CXCL1/2-CXCR2 pair drives neutrophil mobilization into the peripheral blood, while CXCL12-CXCR4 serves as a retention signal that prevents neutrophil egress from the BM [7,8]. Circulating neutrophils include two main populations: (i) mature, normal-density neutrophils that are short-lived cells and provide rapid antimicrobial defense, but also contribute to sterile inflammation after tissue injury, and (ii) low-density neutrophils—often immature or activated forms that co-purify with peripheral blood mononuclear cells after density centrifugation—exhibit distinct immunomodulatory properties. The effector functions of mature neutrophils include phagocytosis, generation of reactive oxygen species (ROS), degranulation, and neutrophil extracellular traps (NETs) release to trap and eliminate microbes [9]. Low-density neutrophils have immunosuppressive activity in cancer and chronic inflammation, and are also involved in promoting thrombosis, tissue damage, and maladaptive responses during infection and autoimmunity [10].

Traditional views hold that neutrophils are absent in normal tissues, although recent research brings evidence that neutrophils actively infiltrate and reside in various tissues, including the heart, even in the absence of infection or injury [11]. In pathological conditions, such as following MI, the DAMPs released from the necrotic area disrupt CXCL12-CXCR4 signaling [12], increase neutrophil mobilization to the peripheral blood, and produce neutrophilia. This disruption can occur through various mechanisms, including direct interference with CXCR4 binding to CXCL12 or by modulating downstream signaling pathways. Neutrophils released from the BM are attracted to the damaged myocardium by cell debris and inflammatory signals released from locally activated resident cells, including cardiomyocytes, cardiac fibroblasts, endothelial cells, and macrophages [13]. Upon infiltration, neutrophils degranulate and release ROS, myeloperoxidase (MPO), cytokines, different proteases (cathepsins, neutrophil elastase, MMP-9, etc.), and form NETs and extracellular vesicles (EVs), actions that both remove necrotic debris and amplify local damage by injuring cardiomyocytes, degrading extracellular matrix, and promoting microvascular obstruction [14]. NETs and neutrophil proteases also promote thrombosis and fibroblast activation, worsening remodeling post-MI [15]. In addition, neutrophils are a major source of alarmins, such as S100A8/A9, which act locally and systemically to amplify granulopoiesis, and of cytokines, that further amplify the inflammatory response by recruiting additional neutrophils and monocytes, activating endothelial cells, and priming resident immune and stromal cells. This pro-inflammatory milieu not only drives the expansion of tissue injury but also shapes the subsequent reparative phase, as signals derived from different neutrophil subtypes critically influence macrophage polarization and the resolution of inflammation [16]. The first study that identified the existence of neutrophil subtypes in the infarct region revealed that N1-pro-inflammatory was the predominant neutrophil subtype in the infarcted LV, reaching the highest levels on day 2 post-MI, while N2 anti-inflammatory subtype increased over the course of MI, attaining 15% of neutrophils on day 7 [17]. Recent advances in single-cell RNA sequencing have further enabled detailed mapping of post-MI dynamics, revealing previously unrecognized heterogeneity among neutrophil populations and their context-dependent roles. Two independent single-cell studies revealed functional diversity among MI-recruited neutrophils in vivo, identifying five to six clusters with distinct, time-dependent patterns (detailed in Section 5.2), and showed that neutrophils remain transcriptionally dynamic beyond the immediate injury phase [14,18]. Neutrophil heterogeneity enables distinct functions depending on the temporal and spatial context, resulting in complex interactions with the heart cells during disease progression. These subtypes are challenging to clearly define, isolate, and reproduce in vitro; therefore, their precise functional roles remain difficult to delineate. However, the N1/N2 neutrophil subtypes were able to be reproduced in vitro [17], and using this model, our group revealed important transcriptomic and functional differences between N1 and N2 neutrophils. We found that N1 neutrophils exhibit higher levels of ROS and oxidative burst (due to elevated expression of NADPH oxidase subunits), increased enzymatic activity of MPO and MMP-9, and enhanced transmigration capacity, partially dependent on autocrine S100A9 signaling [19]. Further studies, which will identify specific markers for each neutrophil subtype identified post-MI, will allow a detailed investigation of their specific functions.

## 3. Role and Diversity of Macrophages in Myocardial Infarction

Monocytes/macrophages are a key component of the innate immune system, and play significant roles in both homeostasis and disease. In healthy adults, resident macrophages are strategically located within the body, where they perform tissue-specific functions and support homeostasis by removing senescent cells/waste and engaging in immune surveillance [20]. These tissue-resident macrophages (TRMs) are mainly established during development and are maintained with minimal or no contribution from circulating monocytes under steady-state [21]. However, following acute injury, large numbers of monocyte-derived macrophages infiltrate the affected tissue and join TRMs in orchestrating both the inflammatory and reparatory response. In the adult mouse heart, TRMs represent around 6–8% of non-myocytes and are found in close association with cardiomyocytes [22]. Based on their origin, two important cardiac TRMs subsets can be distinguished: (1) tissue-resident CCR2^−^ macrophages, which seed the heart during embryonic development, and persist into adulthood through self-renewal, and (2) tissue-resident CCR2^+^ macrophages, which are recruited and replenished from circulating monocytes in a CCR2-dependent manner after birth [23,24].

Macrophage origin is a critical determinant of their function within tissues. As such, embryonic CCR2^−^ cardiac TRMs, which are involved in tissue remodeling and regeneration during development, take on mostly anti-inflammatory roles into adulthood, involving immune surveillance and tissue repair [25]. In contrast, CCR2^+^ TRMs were shown to possess a pro-inflammatory gene signature and are a key driver of the inflammatory response after injury [4].

Following MI, the massive local release of DAMPs by necrotic cells activates resident macrophages through their PRRs, initiating the transcription and release of pro-inflammatory cytokines and chemokines, including CCL2/CCL7 [4]. In response, large numbers of monocytes egress from the BM and extramedullary sites of haematopoiesis and are recruited to the injured myocardium. In mice, the first wave of recruited monocytes consists of the classical Ly6-C^hi^ subset, which dominates the inflammatory phase and peaks on day 3 post-MI [26]. These pro-inflammatory monocytes are recruited in a CCR2-dependent manner [27], and correspond to human CD14^++^CD16^−^ monocytes [28], giving rise to M1-like (classical activated) macrophages following tissue infiltration. M1 macrophages are characterized by the production of pro-inflammatory mediators, including cytokines, chemokines, ROS, and proteolytic enzymes, and are mainly responsible for clearing debris and apoptotic cells to prepare the tissue for scar formation [29]. On day 1 post-MI, macrophages display a pro-inflammatory and matrix-degrading phenotype, and adapt to the hypoxic environment by reprogramming their metabolism towards glycolysis. By day 3 post-MI, macrophages start to downregulate many pro-inflammatory genes while also increasing proliferation and phagocytosis [30], shifting the balance towards inflammation resolution. This may be in part due to upregulation of osteopontin expression, which was shown to enhance macrophage phagocytosis and promote reparative fibrosis in vivo [31]. In vitro, neutrophil secreted factors were also shown to enhance macrophage efferocytosis [16], and neutrophil depletion affected macrophage polarization and led to increased fibrosis and impaired heart function after MI [32].

The second wave consists of Ly-6C^low^ monocytes, which are recruited in a CX(3)CR1-dependent manner, and peak at around day 7 post-MI in mice. These anti-inflammatory monocytes are recruited in smaller numbers compared to the first wave, and are considered analogous to the human non-classical CD14^+^CD16^+^ subset [26]. At this time point, M2-like macrophages become the predominant phenotype in the myocardium and drive the reparatory phase by stimulating cell proliferation, angiogenesis, and ECM synthesis through the release of anti-inflammatory cytokines (IL-10) and growth factors (VEGF, TGF-β) that act on neighboring cells [24]. These macrophages were also observed to express genes typically considered fibroblast-specific, including Col1a1 and periostin, suggesting that they directly contribute to scar formation by transitioning to a fibroblast-like phenotype [30].

Although the simplified M1/M2 dichotomy is used here to describe post-MI macrophage phenotypes, accumulating transcriptomic and lineage tracing data show that macrophage heterogeneity in physiological conditions, and especially after MI, is vastly more complex. Therefore, although traditionally viewed as a uniform population of phagocytes, it is now clear that cardiac macrophages are highly diverse, encompassing both resident populations and monocyte-derived subsets that infiltrate the myocardium after injury and differentiate into diverse subtypes (detailed in Section 5.2). Their phenotypic plasticity enables them to adapt to local signals, transitioning from pro-inflammatory and tissue-degrading roles in the acute phase to reparative, pro-angiogenic, and pro-fibrotic functions during the healing process. This temporal and functional heterogeneity is essential for efficient clearance of necrotic debris, resolution of inflammation, scar formation, and prevention of adverse remodeling.

## 4. Neutrophil–Macrophage Crosstalk: Phenotype Modulation

The dynamic interplay between neutrophils and macrophages represents a central axis of immune regulation in cardiovascular disease, particularly in atherosclerosis and MI. Neutrophils are among the first responders to tissue injury, releasing different effector molecules that not only shape the microenvironment but also influence macrophage recruitment, activation, and polarization. Conversely, macrophages, whether tissue-resident or monocyte-derived, provide signals that dictate neutrophil lifespan, migratory behavior, and possibly effector functions. This bidirectional communication determines whether inflammation is resolved or progresses, and thus has critical implications for tissue repair, fibrosis, and long-term cardiac remodeling.

### 4.1. Temporal Dynamics of Neutrophil–Macrophage Communication Post-MI

The initial inflammatory phase of myocardial infarction (day 1–3) represents the clearest window in which monocyte–neutrophil crosstalk occurs, either directly or indirectly. DAMPs released from ischemic tissue activate PRRs on surviving cells, including resident CCR2^+^ macrophages [5]. These macrophages, in turn, secrete inflammatory cytokines and chemokines that recruit circulating leukocytes, including neutrophils, marking the first level of neutrophil–macrophage communication (Figure 1). Infiltrating neutrophils peak around day 1 post-MI [33], and produce chemokines that signal monocytes to migrate to the lesion site (a second level of communication). Under the influence of local inflammatory cues (many of which are provided by neutrophils), monocytes differentiate into CCR2^+^Ly6C^high^ pro-inflammatory macrophages, a third level of cross-talk. These macrophages actively phagocytose cellular debris, including apoptotic neutrophils, and further amplifying inflammation through the secretion of IL-1β, IL-6, and TNF-α. While neutrophils release inflammatory mediators that exacerbate tissue damage and promote additional immune-cell recruitment [34], they also aid debris clearance and are required for efficient macrophage efferocytosis and polarization toward an M2-like reparative phenotype [35], highlighting that neutrophil–macrophage interaction may participate in the initiation of the reparative phase post-MI, marking a fourth level of interaction (Figure 1). This phase involves the gradual resolution of inflammation, myofibroblast proliferation, and scar formation, typically starting from day 4 to day 7 post-MI. After the first 3 days, anti-inflammatory mediators increase, limiting neutrophil infiltration, enhancing macrophage efferocytosis of apoptotic neutrophils, and possibly promoting the transition of CCR2^+^Ly6C^high^ recruited macrophages toward reparative phenotypes [24]. There is evidence showing that NGAL (neutrophil gelatinase-associated lipocalin) from the neutrophil secretome modulates efferocytosis by regulating the expression of efferocytosis receptor MertK (myeloid-epithelial-reproductive tyrosine kinase) on the membrane of cardiac macrophages [32]. In addition, DNA from NETs can prime Mertk^−^MHC-II^lo-int^ macrophage polarization through the TLR9 pathway, although the precise mechanisms remain to be defined [36]. Both examples illustrate an indirect mode of communication, whereby neutrophil-derived factors modulate macrophage functionality.

Resolution of the inflammatory phase marks the transition to the proliferative phase, a process characterized by angiogenesis and collagen deposition by myofibroblasts, ultimately leading to the development of scar tissue. During this stage, inflammatory leukocytes are either cleared from the injured tissue or transformed into reparative phenotypes that secrete anti-inflammatory, pro-angiogenic, and pro-fibrotic mediators [37]. Collectively, current evidence indicates that neutrophil–macrophage cross-talk occurs predominantly during the inflammatory and early reparative phases of MI, while much less is known about their interactions during the later proliferative phase.

### 4.2. Transcriptomic Signatures of Neutrophil and Macrophage Subsets Post-MI

Macrophages were initially categorized into polarized M1/M2 states; by analogy, N1/N2 neutrophils were proposed in the infarct area based on the gene expression profiling of isolated cardiac neutrophils using pro- and anti-inflammatory markers. In this manner, it was shown that pro-inflammatory molecules (e.g., *IL-1β*, *TNF-α*, *CCL3*) peak on day 1 post-MI in isolated neutrophils, whereas anti-inflammatory markers increase around days 4–7 [17]. It is now clear, however, that both macrophages and neutrophils exhibit broad phenotypic heterogeneity beyond these binary definitions. Recent single-cell RNA-seq (scRNA-seq) studies have revealed transcriptionally distinct immune subpopulations and their dynamic flux after MI [38]. Because neutrophils are particularly sensitive to scRNA-seq workflows, many cardiac datasets under-capture their diversity, and instead predominantly describe the monocyte/macrophage continuum, but emerging studies are overcoming these limitations. Defining these subpopulations and their temporal trajectories could also shed light on the neutrophil–macrophage crosstalk and its influence on the progression versus resolution of inflammation.

A scRNA-seq study [18] mapping neutrophils on days 1, 2, or 3 post-MI identified five cardiac neutrophil states: *Retnlg^HI^*, *ISG^HI^*, *NF-κB^HI^*, *HIF-1^HI^*, and *SiglecF^HI^*. *Retnlg^HI^* neutrophils were the most abundant on day 1 and were also prominent in the steady-state heart. Notably, whereas other subsets declined by days 2 and 4, the *SiglecF^HI^* population increased and reached ~54% of all cardiac neutrophils by day 4. These findings are supported by another study, which reported a similar *SiglecF^H^*^I^ neutrophil subset emerging from day 3 post-MI onward [14]. This *SiglecF^HI^* population exhibits a pro-inflammatory phenotype with enriched *NF-κB* and *Myc* target activity and elevated expression of genes such as *TNF*, *ICAM-1*, *Nlrp3*, features consistent with enhanced longevity and sustained inflammatory signaling. By comparing aging and transcriptomic specifications in circulating versus myocardial neutrophils, these authors also showed that neutrophils can acquire specialized phenotypes locally within infarcted tissue. Both studies further identified a type-I interferon-responsive subset (*ISG^HI^*) present in both normal tissue and blood, indicating that this state is not MI-specific [39].

Similar to neutrophils, macrophages can also assume multiple fates in the infarcted heart. One important distinction must be made between the recruited versus resident macrophages. CCR2 status was initially used as a marker of residency, with CCR2^+^ macrophages being considered monocyte-derived and CCR2^−^ cells viewed as seeding the heart during embryonic development [4]. However, scRNA-seq is revealing that some monocyte-derived macrophages express low CCR2, indicating that additional markers are required to discriminate these lineages. In a comprehensive 2022 study, Dick et al. showed that three resident subsets co-exist in the normal heart: a TLF^+^ population expressing TIMD4, LYVE1, FOLR2 with low CCR2 and self-renewal capacity; a CCR2^+^ subset with low TIMD4^−^LYVE1^−^FOLR2^−^, which is constantly replaced by monocytes; and a MHC-II^hi^ subset that lacked TLF subset genes and CCR2 expression [40]. A recent integrative study showed a macrophage cluster with high *Lyve1* and *Folr2* expression that was enriched in controls, declined rapidly by day 3 post-MI, and was again increased in the reparative phase after day 7 [41]. Chakarov et al. identified a similar macrophage population high in *Lyve1* expression that was localized near blood vessels and had a potential role in facilitating vascular repair and regeneration through the release of vascular growth factors in response to injury [42]. The modulation of this cluster post-MI could prove useful in future therapeutic actions post-MI, as there is evidence that human Lyve^+^ macrophages can also interact with cardiac fibroblasts and influence cardiac fibrosis through the CD74-MIF axis [43].

For infiltrating macrophages, Ke et al. integrated multiple scRNA-seq and snRNA-seq datasets and resolved eight macrophage subclusters in mouse hearts [41]. One cluster with high *Arg1*, *Ccl6*, and *Ccl2* expression expanded rapidly on day 1 post-MI and declined by day 5. In contrast, a *Spp1^+^*, *Gpnmb^+^*, *Fabp5^+^* cluster expanded from day 3, peaked on day 7, and dropped sharply by day 14. In both mouse and human cardiac tissues, *Spp1^high^* macrophages were almost exclusively present in early MI, not in later stages. Arg1 is a canonical marker for immunosuppressive M2 macrophages [44]; in postnatal day 1 (P1) vs. P10 mice hearts, myocardial injury was associated with an increased *Arg1^+^* macrophage subpopulation in regenerative P1 hearts, correlating with improved repair. Interestingly, CXCR2 inhibition enhanced cardiac repair and further increased Arg1^+^ macrophages, but this effect was lost when neutrophils were depleted [45], underscoring the important role of neutrophils in post-MI repair.

While defining the transcriptomic signatures of neutrophil and macrophage subsets is essential, capturing their communication potential is paramount. scRNA-seq now enable inference of cell–cell interactions using tools such as Scriabin, NicheNet, LIANA+, Cellchat [46]. Applying NicheNet to neutrophil–macrophage signaling, Vafadarnejad et al. identified *IL-1β, TNF-α*, and *Csf1* as top-ranked neutrophil ligands influencing macrophage programs, and highlighted Csf1/Csf1r and Vegfa/Nrp1 as key ligand-receptor pairs in this interaction. In addition, some ligands exhibited enrichment in neutrophils at different time points after MI, for example, TNF-α peaking on days 3 and 5 post-MI, and C3 and Ccl4 on day 1 post-MI, suggesting a time-dependent neutrophil/macrophage cross-talk [14].

Although scRNA-seq provides valuable insights into cellular transcriptomic heterogeneity and enables the inference of cell–cell communication, the required tissue dissociation results in the loss of spatial context. To overcome this limitation, spatial transcriptomics (ST) technologies preserve cellular localization and spatial relationships of neighboring cells. A recent study employing ST revealed that neutrophils highly infiltrate into the infarcted area on day 1 post-MI, whereas macrophages are mostly dispersed across the whole heart tissue. Furthermore, while early (days 1–3) and late (days 5 and 7 post-MI) macrophage subsets accumulate in the infarcted region, a subset of steady-state tissue resident macrophages remains dispersed across the myocardium at all time points, potentially modulating the immune response and promoting tissue repair [47].

Another comprehensive study employing both scRNA-seq and ST that focused on the cellular composition and distribution within the MI border zone revealed the presence of several myeloid subsets, including *Lyve1*^HI^ resident macrophages, monocyte-derived macrophages (*Ly6c2*^HI^, *Ifit1*^HI^, *Arg1*^HI^, *C1qa*^HI^, *Gclm*^HI^), and neutrophils (*Retnlg^HI^, Siglecf^HI^, Ifit1^HI^*) in the first 7 days post-MI. Macrophage subpopulations (*C1qa*^HI^, *Gclm*^HI^, *Lyve1*^HI^) were located along the posterolateral wall of the LV, near the papillary muscle, while *Arg1*^HI^ pro-resolving macrophages were found in the anterior and anterolateral wall of the LV. Neutrophil markers (*S100a8*, *S100a9*) and fibrosis were also abundantly detected in this area, which is the most affected following the occlusion of the coronary artery. This data supports the potential for local interactions between the two cell populations and provides the necessary context for neutrophils to promote a pro-resolving macrophage phenotype. In addition, both *Ifit1*^HI^ macrophages and *Ifit1*^HI^ neutrophil subsets, which correspond to a mostly pro-inflammatory signature, were found to inhabit the same limited area in the anterior wall of the LV [48].

Therefore, despite major advances from time-resolved single-cell sequencing, we still lack a causally resolved, time-anchored map of how discrete neutrophil and macrophage subtypes communicate across infarct zones to govern the transition from inflammation to repair. Integrative frameworks for inferring cell–cell communication now exist, but they require further studies to move beyond correlation and establish signaling mechanisms. Systematic, phase-specific mapping of who talks to whom, when, and where will be essential to explain why similar cellular compositions can yield different remodeling trajectories after MI.

### 4.3. Factors Produced by Neutrophils as Modulators of Macrophage Phenotype

The neutrophil secretome—including cytokines, proteases, NETs, and extracellular vesicles—can influence the behavior of infiltrating monocytes and macrophages, controlling their polarization and function (Table 1). The factors released by neutrophils expand the inflammatory infiltrate but also create a microenvironmental context that primes macrophages either for pro-inflammatory or reparative functions, depending on the timing and balance of additional signals. Our recent data showed that macrophages exposed to the secretome from pro-inflammatory N1 neutrophils exhibit increased expression of *TNF-α*, *IL-1β*, and *S100A8/A9*, suggesting that factors released by these cells promote an M1-like inflammatory response in macrophages. In contrast, when macrophages were exposed to N2 secretome, they increased expression of the anti-inflammatory markers *CD206*, *TGF-β*, and *IL-10* and of nuclear factors associated with reparatory macrophages (PPARγ, Nur77, and KLF4), indicating that factors released by N2 cells promote an M2-anti-inflammatory response [16]. Moreover, MerTK—an indispensable molecule for clearance of cell debris and inflammation resolution post-MI—together with other efferocytosis receptors were upregulated in macrophages following their exposure to soluble factors released by anti-inflammatory N2 neutrophils [16], highlighting the importance of neutrophil–macrophage cross-talk in promoting inflammation resolution. Therefore, this secretome-driven crosstalk plays a central role in the transition from inflammation to repair, potentially affecting processes such as macrophage recruitment, efferocytosis, and fibrotic remodeling. Among the factors contained in the neutrophil secretome that impact macrophage phenotype and functionality, there are as follows:

#### 4.3.1. Cytokines/Chemokines

Neutrophils express and produce cytokines and chemokines either constitutively or upon activation by microenvironmental stimuli [49]. After their recruitment to the ischemic myocardium, neutrophils secrete a wide range of cytokines and chemokines that act in a paracrine manner to shape macrophage responses. Therefore, upon activation, neutrophils secrete CXCL1 and CXCL8 (IL-8), which act in a paracrine manner to attract circulating monocytes and enhance their extravasation through endothelial barriers, but also CCL3 and CCL4, which contribute to the recruitment of CCR5^+^ monocytes and influence their differentiation toward pro-inflammatory macrophages once within the ischemic tissue [50,51]. Beyond recruitment, neutrophil-derived chemokines may also dictate the spatial distribution of macrophages, directing them toward necrotic cores or border zones where clearance and remodeling are most needed.

Besides chemokines, neutrophils secrete an important number of pro-inflammatory cytokines such as TNF-α, IL-6, and IL-1β. These cytokines activate surrounding endothelial and stromal cells, amplifying chemokine gradients (such as CCL2) that further attract monocytes with a pro-inflammatory bias [52]. In addition, they polarize infiltrating macrophages toward an M1-like phenotype, characterized by production of ROS, proteases, and inflammatory cytokines, necessary in the initial days post-MI for efficient clearance of necrotic debris. If this signal persists or is exacerbated, it can induce tissue damage, the overall effect becoming damaging. IFNγ—a pro-inflammatory cytokine and immunomodulatory molecule also produced by neutrophils—influences the overall immune response and the behavior of other immune cells, including macrophage activation towards M1 macrophages through signaling pathways like JAK-STAT1. It can also induce epigenetic changes and metabolic reprogramming to alter the cell’s response to other stimuli [53], playing a central role in maintaining trained immunity in vivo and inducing potent memory in macrophages [54].

Pro-inflammatory and immunoregulatory IL-17 is another cytokine secreted by neutrophils, which can modulate the activity of immune cells, including macrophages. Macrophages exposed to IL-17 do not differentiate into classical M1 or M2 subsets but instead upregulate innate immune receptors, TLR2 and TLR4 [55], thereby becoming more responsive to diverse DAMPs/PAMPs and other agonists that signal using TLRs.

In parallel, neutrophils can also release anti-inflammatory mediators, which act on macrophages to suppress NF-κB activation, promote efferocytosis, and stimulate their polarization toward a reparative M2-like state [16,56]. Notably, IL-10 acts as a potent anti-inflammatory cytokine, significantly inhibiting macrophage activation, reducing ROS, and suppressing the release of cytokines like TNF-α, IL-1β, and IL-6 [57]. It also alters macrophage metabolism by inhibiting inflammation-induced glycolysis and promoting oxidative phosphorylation and autophagy [58].

#### 4.3.2. Granule Enzymes

Neutrophils release a variety of granule-stored enzymes and proteins that exert paracrine effects on macrophages in the infarcted myocardium. MPO, an important neutrophil enzyme, unique in its ability to generate reactive chlorinating species such as hypochlorous acid (HOCl), possesses potent bactericidal and viricidal activities, having an essential role in the innate immune response [59]. While HOCl formation is important in pathogen removal, it is also involved in host tissue damage and multiple inflammatory diseases. MPO has numerous inflammatory properties, and MPO plasma levels are correlated with prognosis and severity of MI [60]. Therefore, both MPO and its reactive oxidants have been implicated in myocardial infarction, which contributes to matrix degradation and oxidative stress [59], and modulate macrophage activation toward a pro-inflammatory phenotype [61]. Post-MI, MPO induces monocyte migration by upregulation of the chemokine receptor CCR2 and of inflammatory signaling pathways [59,60].

Similarly, neutrophil elastase and cathepsin G are two neutrophil serine proteases that degrade extracellular matrix components like elastin and fibronectin, as well as surface receptors [62]. This degradation releases bioactive fragments, known as matrikines, that enhance macrophage recruitment and sustain inflammatory signaling [63]. In addition, neutrophil-derived matrix metalloproteinases (MMPs), particularly MMP-8 and MMP-9, remodel the extracellular matrix and facilitate monocyte/macrophage infiltration into the injured tissue [64]. These enzymes also liberate matrix-bound growth factors and cytokines that indirectly influence macrophage polarization toward either inflammatory or reparative phenotypes, depending on the local environment [65]. Therefore, neutrophil granule enzymes function as molecular amplifiers of inflammation but also as modulators of repair, reinforcing the dual role of neutrophils as both initiators of damage and orchestrators of healing through their impact on macrophage behavior.

#### 4.3.3. DAMPs and TLR Signaling

Among the most important DAMPs released by neutrophils upon infiltration into injured cardiac tissue, there are high-mobility group box 1 (HMGB1) and S100 proteins [66,67]. These molecules function as innate amplifiers of inflammation, modulating the local immune environment. Thus, neutrophil interactions with cardiac macrophages are also mediated by the DAMPs released under ischemic stress [68]. Once in the extracellular compartment, they bind to PRRs, such as TLRs, present on macrophages, with TLR4 being one of the most potent signal receptors associated with inflammatory responses [69]. HMGB1—an evolutionarily conserved and ubiquitously expressed protein considered to be the prototypical DAMP—is typically a nuclear protein, released either passively from necrotic neutrophils or actively secreted during degranulation and NET formation [67]. As a key neutrophil-derived DAMP, it activates macrophages, driving a pro-inflammatory program via activating the NF-kB pathways through its interaction with TLR4 [69].

Similarly, S100 proteins are abundant in neutrophil cytosol and are released during activation or cell death. Neutrophils are the major source of S100A8/A9, a key alarmin responsible for the immunostimulatory effects in MI, which induces immune cell infiltration and activates the inflammasome priming in macrophages via TLR4/RAGE engagement [70]. The alarmin is rapidly released from neutrophils upon their activation to modulate the production of IL-1β in an autocrine manner, or in macrophages in a paracrine fashion. Moreover, it promotes the expression of multiple pro-inflammatory mediators, including IL-6, IL-8, and TNF-α, and induces ROS production [68]. Besides its role in promoting a pro-inflammatory phenotype in macrophages within the first 1–3 days post-MI, overwhelming data support a cardioprotective, regulatory function of S100A8/A9 by inducing a reparatory macrophage phenotype on day 7 post-MI [71]. Importantly, short-term blockade of S100A9 using the specific blocker ABR-238901 during the early inflammatory phase inhibits inflammation and improves cardiac function, while long-term blockade has negative effects, impairing monocyte recruitment and the differentiation of monocytes into reparative macrophages [71], which are essential for healing. Thus, mice receiving prolonged S100A9 blockade failed to switch to M2 phenotype due to a pronounced downregulation of MerTK, causing accumulation of apoptotic cells and subsequent adverse cardiac remodeling, followed by a decline in cardiac function [66]. The impaired transition from Ly6Chi to Ly6Clo macrophages was at least in part due to Nur77 inhibition by the S100A9 inhibitor, with Nur77 being an orphan nuclear receptor that orchestrates M2 polarization. Nonetheless, prolonged neutrophil-derived alarmin inhibition was also responsible for the diminished monocyte mobilization in the ischemic milieu of the myocardium, resulting in an even larger decrease in Ly6CloMerTKhi macrophages [66].

However, when discussing DAMPs and their impact on macrophages in MI, it is important to note that, in addition to HMGB1 and S100 proteins, several other DAMPs contribute to shaping macrophage phenotype and behavior. These include heat-shock proteins (HSP60, HSP72), mitochondrial peptides, and extracellular nucleic acids (mtDNA, RNA), which are released not only by neutrophils but also from the necrotic tissue microenvironment [72].

#### 4.3.4. Neutrophil Extracellular Traps

NETs, composed of extracellular DNA decorated with histones and neutrophil-derived proteins such as MPO and elastase, are increasingly recognized as critical paracrine modulators of macrophage function after myocardial infarction. Post-MI, NETs have a dual role in influencing macrophage behavior, potentially promoting both pro-inflammatory responses in the early phase [73] that hinder healing, and reparative phenotypes that support scar formation and tissue remodeling in the latter phase [74]. NETs can act as a pro-inflammatory stimulus, as extracellular histones and DNA fragments activate PRRs on macrophages, triggering the NLRP3 inflammasome, sustaining cytokine release [75]. NET-associated proteases can further amplify this inflammatory state by degrading the extracellular matrix, releasing DAMPs, and perpetuating leukocyte recruitment. Thus, NETs can induce macrophages to polarize in M1 phenotype in gouty inflammation via targeting hexokinase-2 [76,77], but if macrophages successfully clear or break down NETs, this action may reduce the stimulus for M1 polarization, thereby allowing macrophages to shift towards an M2 phenotype. In this line, it was found that NETs also promote the M2 reparative macrophage phenotype and IL-10 anti-inflammatory cytokine expression, facilitating wound healing and fibrosis [74]. Macrophages play a central role in clearing NETs through phagocytosis of extracellular DNA and associated proteins, and MMP12 was identified as a major mediator of macrophage-driven NET clearance [78]. Although there is some evidence suggesting that efficient NET clearance dampens inflammatory pathways to prepare for the resolution of inflammation [74,78], future research should elucidate how macrophage-mediated NET degradation influences the resolution of inflammation post-MI. Thus, NETs embody the dual nature of neutrophil–macrophage crosstalk: they can either perpetuate inflammatory injury or, when efficiently cleared, contribute to the transition toward repair. The overall effect depends on the balance of these actions and the stage of the healing process.

#### 4.3.5. Extracellular Vesicles

Neutrophils release extracellular vesicles (EVs), including microvesicles and exosomes, as a means of paracrine communication with macrophages and other cardiac cells. These vesicles carry a diverse cargo of proteins, lipids, nucleic acids, and enzymes that reflect the activation state of the parent neutrophil [79]. Although extracellular vesicles are known to influence macrophage phenotypes by transferring bioactive cargo such as microRNAs, alarmins, and inflammatory mediators [80], the specific contribution of neutrophil-derived EVs to macrophage modulation has been poorly investigated. Until now, it has been shown that neutrophil microvesicles prevent inflammatory activation of macrophages and promote the release of TGF-β [81]. Similarly, a recent study revealed that neutrophil-derived EVs act as mediators of neutrophil–macrophage communication, modulating itaconate metabolism in macrophages and thereby protecting against hyperinflammation in cytokine storm syndrome [82]. These results were the first to suggest that concomitant with the release of powerful pro-inflammatory mediators, neutrophils also produce and release EVs with immunosuppressive function that can down-regulate the inflammatory response. Annexin A1 (AnxA1) is one of the pro-resolving proteins released in neutrophil EVs following inflammatory stimuli, and can trigger anti-inflammatory pathways by binding to its receptors FPR1 and FPR2, expressed mostly by phagocytic cells [83]. Recent studies showed that AnxA1 counteracts classical activation of macrophages and promotes the release of TGF-β both in vitro and in vivo [81]. Furthermore, AnxA1 promoted neovascularization and cardiac repair in a mouse model of MI by stimulating cardiac macrophages toward a pro-angiogenic phenotype that releases VEGF-A [84]. Importantly, the dual pro-inflammatory or pro-resolving effects of neutrophil-derived EVs are determined by their cargo composition, which varies based on the nature of the activating stimulus and the functional state of neutrophils at the time of EV release [85].

**Table 1 ijms-26-10575-t001:** Molecular mediators produced that modulate monocyte (Mon), macrophage (Mac) or neutrophil (N) behavior/phenotype.

Molecular Mediators	Source/Target Cell	Impact on the Target Cell	Reference
CXCL1, CXCL8, CCL3, CCL4	N/Mac	monocyte chemotaxis and adhesion, M1 macrophage	[50,51]
TNF-α, IL-6, IL-1β	N/Mon, ECs	attract pro-inflammatory Mon, induce M1-Mac, activate ECs	[52]
IFNγ	N/Mac	induces M1-Mac, epigenetic changes and metabolic reprogramming	[53]
IL-17	N/Mac	upregulate innate immune receptors, TLR2 and TLR4	[55]
IL-10	N/Mac	↓ macrophage activation, inhibits glycolysis and promotes oxidative phosphorylation and autophagy, suppressing the release of cytokines	[57,58]
MPO	N/Mon	↑ CCR2 and monocyte migration, induces M1-Mac	[59,60,61]
elastase and cathepsin G	N/Mon	enhances macrophage recruitment through release of matrikines, sustains inflammatory signaling	[63]
MMP8 and MMP-9	N/Mon, Mac	monocyte/macrophage infiltration; indirect macrophage M1/M2 polarization	[65]
HMGB1	N/Mac	NF-kB pathway activation via TLR4	[69]
S100A8/A9	N/Mac, N	acute MI: ↑IL-1β, IL-6, IL-8, and TNF-α, and induces ROS. day 7 post-MI: M2-Mac polarization	[68,70,71]
NETs	N/Mac	NLRP3 inflammasome activation, cytokine and DAMP release, leukocyte recruitment, M1/M2 polarization	[74,75,76,77]
MMP12	Mac/N	NETs clearance	[78]
EVs (AnxA1)	N/Mac	counteracts M1-Mac activation, promotes TGF-β and VEGF-A release	[81]

### 4.4. Role of Macrophages as Modulators of Neutrophil Phenotypes

As stated, following MI, neutrophils are the first immune cells to infiltrate the injured myocardium, followed by monocytes. Most studies emphasize that infiltrating monocytes are modulated by neutrophils already present within the lesion, suggesting that the earliest cross-talk between these populations is directed from neutrophils toward monocytes/macrophages. Despite this, in the very early phase of MI, resident macrophages within the infarct zone are rapidly activated by local DAMPs and release chemokines that attract neutrophils. Thus, the first wave of crosstalk is actually directed from macrophages toward neutrophils. Therefore, it can be argued that resident macrophages play an initiating role in post-MI evolution by recruiting neutrophils to the lesion.

As reviewed in the previous section, many studies have investigated how neutrophils, or their released factors, influence macrophage phenotypes and functions. In contrast, only a limited number of studies have examined the reciprocal process—how macrophages affect neutrophil behavior. A recent study addressed this gap, showing that conditioned medium from activated human monocyte-derived macrophages enhances neutrophil activation by boosting both oxidative and glycolytic metabolism, while simultaneously impairing their migratory capacity, effectively retaining them at sites of injury. These neutrophils displayed elevated ROS production, increased NET formation, and enhanced secretion of CXCL8, IL-13, and IL-6 compared to untreated neutrophils or those exposed to conditioned medium from unstimulated macrophages [86].

### 4.5. Metabolic Crosstalk Between Neutrophils and Macrophages

The interplay between neutrophils and macrophages extends beyond cytokine and EV signaling to include a metabolic crosstalk within the infarcted tissue microenvironment. Cellular metabolism is a key determinant of immune cell function and plasticity, as demonstrated in T cells and macrophages [87]. In contrast, the metabolic regulation of neutrophil biology has only recently gained attention. Traditionally regarded as cells relying almost exclusively on glycolysis, neutrophils were thought to possess limited metabolic flexibility. However, recent advances have questioned this view, revealing that the tricarboxylic acid cycle, oxidative phosphorylation, fatty acid oxidation, and the pentose phosphate pathway all contribute to meet the energetic, biosynthetic, and functional demands of neutrophils [88,89,90,91]. Although the metabolic interplay between neutrophils and macrophages following MI has not yet been extensively investigated, emerging evidence suggests the existence of such a crosstalk. Recent data indicate that conditioned media from activated macrophages enhances neutrophil activation, as reflected by increased ROS production, elevated NET formation, and higher levels of CXCL8, IL-13, and IL-6. These changes were accompanied by an upregulation of both oxidative and glycolytic metabolism, while concurrently reducing neutrophil migratory capacity [86]. Moreover, metabolic intermediates produced by these two cell types, such as lactate, succinate, and arginine-derived metabolites, can profoundly influence neutrophil and macrophage polarization and behavior by modulating their activation, lifespan, and effector functions. Therefore, the acute inflammatory response promotes an increase in glycolytic activity within neutrophils, leading to an increase in lactate production and release [92]. Lactate produced during neutrophil glycolysis stimulate the expression of pro-inflammatory genes in the early stage of inflammation [93]. However, as lactate accumulates, the process of lactylation triggers M2-like macrophage polarization in a time-dependent manner, referred to as the ‘lactate clock’, thereby regulating the inflammatory response. In the late phase of inflammation, lactylation induces the transformation of M1-like macrophages into M2-like macrophages through epigenetic mechanisms, thus facilitating the repair of tissue damaged caused by inflammation [94].

Succinate has also been shown to exert different immunomodulatory effects in macrophages. During LPS-induced activation, macrophages shift from oxidative phosphorylation to glycolysis, leading to intracellular succinate accumulation that stabilizes HIF-1α and promotes IL-1β production, reinforcing M1 polarization [95]. Conversely, extracellular succinate can signal through the SUCNR1 receptor to favor M2-like polarization, thereby contributing to an immunosuppressive tissue environment [96].

Arginine is a key amino acid for macrophages, and its metabolism determines their function, particularly in the context of macrophage polarization. Upon activation, macrophages use L-arginine through two major competing enzymatic pathways: nitric oxide synthase (iNOS) and arginase. In classically activated M1 macrophages, inducible iNOS converts arginine to nitric oxide (NO) and citrulline, promoting antimicrobial activity and sustaining inflammatory responses [97]. In contrast, alternatively activated M2 macrophages preferentially express arginase-1 (Arg1), which metabolizes arginine into ornithine and urea. Ornithine serves as a precursor for polyamines and proline, facilitating tissue repair, collagen synthesis, and wound healing [97]. Activated neutrophils release Arg1 from cytoplasmic granules, thereby competing with macrophage iNOS for the shared substrate L-arginine [98]. Arg1 was significantly up-regulated in different MI animal models, and arginase activity was demonstrated to be markedly up-regulated as early as 20 min after reperfusion and maintained on day 8 post-MI [99,100]. Increased arginase competes with iNOS for arginine utilization, resulting in decreased NO production and citrulline/ornithine ratio [99,101]. Depletion of local arginine availability suppressed NO production in macrophages, attenuated pro-inflammatory M1 polarization [102], and promoted a shift toward a reparative, M2-like phenotype [103]. Additionally, immature neutrophils, which are released during the inflammatory phase [104] generate increased levels of arginine-derived metabolites like ornithine and spermidine [105]. Ornithine was shown to modulate macrophage function, particularly through the enzyme ornithine decarboxylase, which regulates M1 macrophage activation during infection [106]. In some contexts, ornithine from arginine metabolism is converted into polyamines like putrescine, which is necessary for optimal macrophage responses, such as the internalization of apoptotic cells [107]. Spermidine has been shown to promote an anti-inflammatory M2-like phenotype through mtROS-dependent AMPK activation, Hif-1α stabilization and autophagy induction [108].

All these findings support the concept of a reciprocal metabolic dialog between neutrophils and macrophages. Through their released metabolites and metabolic enzymes, each cell type adjusts the other’s energy metabolism, polarization state, and inflammatory output, thereby influencing the delicate balance between inflammation and repair in cardiovascular injury.

## 5. Pathological Consequences of Dysregulated Immune Interaction

The coordinated interaction between neutrophils and macrophages is essential for efficient host defense, clearance of necrotic debris, and initiation of tissue repair. However, when this crosstalk becomes dysregulated due to: excessive activation, persistent inflammation in the early stage, age- and sex-related immune response differences, or maladaptive signaling, it can drive pathological inflammation. In the cardiovascular system, such imbalances contribute to adverse outcomes after myocardial infarction, including excessive tissue remodeling, impaired resolution of inflammation, and progressive fibrosis that can lead to heart failure.

### 5.1. Altered Balance of Resident Cardiac Macrophage Subsets Affects Early Neutrophil Recruitment Following MI

Resident CCR2^−^ and CCR2^+^ cardiac macrophages act as the first sensors of released DAMPs following cardiomyocyte ischemic cell death. Recent studies revealed that these two subsets differentially regulate neutrophil and monocyte recruitment after myocardial injury. CCR2^+^ macrophages were found to be indispensable for neutrophil extravasation into the myocardium in a mouse model of ischemia–reperfusion. In this model, free DNA and histones released by necrotic cardiomyocytes served as activators of CCR2^+^ resident cardiac macrophages via the TLR9/MyD88 signaling axis, prompting them to produce and release CXCL2 and CXCL5. Both proteins serve critical roles in neutrophil recruitment, as the absence of CXCL5 was associated with defects in crawling on the endothelial surface, and CXCL2 interrupts the cascade more upstream, by impairing neutrophil adhesion to the endothelium [5]. Also, in a mouse model of MI, depletion of resident CCR2^+^ macrophages resulted in substantially less monocyte, macrophage, and neutrophil recruitment in the first 2 days following myocardial injury, and reduced interstitial fibrosis after 28 days, suggesting that depletion of CCR2^+^ macrophages alone is enough to limit the initial inflammatory response and subsequent fibrosis. In contrast, a lack of resident cardiac CCR2^−^ macrophages led to shifts in monocyte fate specification, increased macrophage proliferation, increased infarct area, reduced LV systolic function, and exaggerated LV remodeling, highlighting their protective role [4].

Collectively, these findings indicate that resident macrophage subsets not only dictate the recruitment of leukocytes to the myocardium but also determine their fate after infiltration. In addition, it points to CCR2^+^ resident cardiac macrophages as an essential upstream mediator of the inflammatory response to cardiac injury, by regulating the intensity of recruitment in the first wave of both neutrophils and monocytes to the affected tissue. As such, disruption of the balance between CCR2^−^ and CCR2^+^ resident cardiac macrophage subsets can either dampen or exacerbate the inflammatory response to injury, with important effects on subsequent tissue repair dynamics and cardiac function. The aging heart provides an eloquent example of the consequences of such an imbalance. Here, the self-renewal capacity of CCR2-resident embryonic cardiac macrophages declines with age, and they are gradually replaced by CCR2^+^ macrophages derived from circulating monocytes [25,109]. RNAseq analysis revealed that recruited CCR2^+^ macrophages are distinct from tissue-resident CCR2^+^ and CCR2^−^ macrophage subsets, and are a particularly inflammatory population that expresses higher levels of inflammatory chemokines (Cxcl1, Cxcl2, Ccl2, Ccl7, Ccl9), cytokines (IL-1β, IL-10), and genes implicated in adverse cardiac remodeling (Areg, Ereg, Gdf3) [4]. As this inflammatory subset becomes the dominant resident macrophage population, the heart becomes primed to initiate an exacerbated inflammatory response when tissue injury occurs, explaining in part the association with excessive inflammation and worse outcomes after MI [17].

Despite accumulating evidence supporting the role of CCR2^+^ and CCR2^−^ resident macrophages in LV remodeling following MI and of more in-depth characterization of the gene profile of these subsets, the mechanisms involved in their interaction and cross-talk with infiltrating neutrophils and monocyte-derived macrophage subtypes following injury remain unclear.

### 5.2. Impaired Inflammation Induced by Altered Neutrophil–Macrophage Cross-Talk

After MI, neutrophils tend to accumulate in the border zone [110], where they perform key functions to prepare the injured tissue for scar formation and ensure macrophage recruitment and polarization. The sequence of inflammatory and reparative events that is triggered following MI serves to ensure scar formation and preserve adequate heart function. For this purpose, the intensity and temporal organization of these events must be tightly regulated, either through direct cell–cell interactions or paracrine signaling. Because of their central role in the inflammatory phase, increased neutrophil infiltration or delayed neutrophil removal can lead to a prolonged inflammatory response and contribute to post-MI adverse LV remodeling. In contrast, in the absence of robust recruitment of neutrophils, debris is improperly removed, resulting in the accumulation of excess granulation tissue and unstable scar formation. A landmark study by Horckmans et al. showed that neutrophil depletion in mice worsens cardiac function and accelerates progression to heart failure after MI, providing direct evidence that neutrophils are indispensable for effective repair [32]. These mice appeared to “skip” the inflammatory phase and showed reduced expression of M1 markers, together with an increase in resident macrophage proliferation. Also, although M2 markers were upregulated in the heart, MertK expression was decreased, and removal of apoptotic cells was impaired, pointing to neutrophils as critical regulators of the microenvironment that drives macrophage polarization towards M1 and its importance for subsequent tissue repair.

Excessive M1 polarization can amplify the inflammatory phase with deleterious effects on tissue repair. This exacerbated inflammation may be due to HMGB1 and S100A8/A9, which signal through TLR4 and RAGE on monocytes/macrophages and neutrophils, driving their differentiation toward pro-inflammatory phenotypes [17,111]. Elevated circulating levels of these DAMPs have been reported in MI patients and were correlated with adverse clinical outcomes such as heart failure, cardiac rupture, and in-hospital cardiac death [112]. These complications likely arise from excessive immune cell recruitment and their polarization into a pro-inflammatory subtype, leading to exacerbated inflammation and impaired resolution. Another mechanism responsible for defective resolution may be dependent on abundant MMPs released from neutrophil granules. Among these, MMP-9 can activate IL-1β and IL-8 by proteolytic cleavage, generating a positive feedback loop for neutrophil and macrophage activation and chemotaxis [113]. This can also result in prolonged inflammation, and MMP-9 plasma levels were correlated with MI mortality, LV remodeling, and dysfunction in humans [114].

Timely apoptosis of activated neutrophils is known to limit inflammation and promote the transition to the reparatory phase after MI. One of the mechanisms responsible for neutrophil apoptosis and removal in the MI microenvironment is driven by MMP-12, a protein mainly secreted by macrophages. In vivo, MMP-12 inhibition resulted in prolonged pro-inflammatory cytokine upregulation by neutrophils, failed upregulation of TGFβ proteins, hyaluronic acid accumulation, and impaired inflammation resolution after MI. Even though inhibition of MMP-12 did not affect the initial size of the infarct area, it resulted in a greater reduction in ejection fraction because of adverse LV remodeling on day 7 post-MI [115]. Moreover, apoptotic neutrophils can release annexin A1 and lactoferrin, which further inhibit neutrophil recruitment and promote infiltration of anti-inflammatory macrophages [116]. Since MMP-12 is preferentially secreted by M2 macrophages, this further establishes a positive feedback loop that promotes inflammation resolution.

### 5.3. Age- and Sex-Driven Immune Dysregulation in Cardiovascular Injury

Sex and age are emerging as critical determinants of immune responses following MI, influencing both the magnitude of inflammation and repair. Experimental and clinical data indicate that male and female hearts exhibit distinct inflammatory trajectories post-MI, partly driven by differences in sex hormones and immune cell composition [117,118]. Older MI patients exhibit ~30% higher mortality compared with younger individuals [119], a difference partly attributed to an exaggerated inflammatory response [120] driven by chronic, low-grade sterile inflammation known as inflammaging.

In aged mice, neutrophil production and recruitment to the infarcted myocardium are diminished [121], whereas in humans, the number remains largely unchanged, but their functionality is impaired, showing delayed apoptosis, increased oxidative bursts, and excessive NET formation [122,123]. Similarly, both human and mouse macrophages exhibit a reduction in MHC-II surface expression [124], with mouse macrophages also exhibiting diminished cytokine production and impaired phagocytic activity [125]. Enhanced neutrophil-driven inflammation activates macrophages, whose polarization also becomes dysregulated with aging. Instead of efficiently transitioning from the pro-inflammatory M1 phenotype toward the reparative M2 phenotype after MI, aged macrophages maintain an inflammatory profile, secreting IL-1β, TNF-α, and IL-6. This dysregulation impairs clearance of necrotic tissue, prolongs inflammation, and reinforces neutrophil–macrophage crosstalk, creating a self-sustaining inflammatory loop. Consequently, these processes promote fibrosis and adverse ventricular remodeling, contributing to impaired cardiac repair, functional decline, and residual inflammatory risk [126,127]. In addition, aging profoundly impacts the adaptive immune response, leading to reduced circulating lymphocyte number and the accumulation of functionally impaired memory lymphocytes [128]. All these changes impact neutrophil–macrophage cross-talk and clinically translate into an elevated neutrophil-to-lymphocyte ratio (NLR) in the elderly, a marker currently used as an indicator of systemic inflammation. Elevated NLR has been shown to correlate with organ damage [129] and to predict increased morbidity and mortality following cardiovascular events [130]. In addition, NLR was shown to be a strong predictor of the presence and number of carotid atherosclerotic plaques in older adults, performing better than CRP and fibrinogen [131], and was positively correlated with the Global Registry of Acute Coronary Events (GRACE) score currently used to predict in-hospital mortality of patients with acute MI [132]. Moreover, NLR can enhance the predictive precision of the GRACE score and can be used independently to forecast the risk of major adverse cardiovascular events (MACE) during hospitalization [133].

Regarding sex and age differences, men tend to experience acute MI at younger ages than women, who typically present about 7-10 years later, often post-menopause. Estrogen, mainly as 17β-estradiol (E2), plays a key role in protecting women’s cardiovascular health. After menopause, E2 levels drop sharply to levels similar to men, which is linked to increased cardiovascular risk and faster LV remodeling [134].

After MI, notable sex-based differences emerge in the innate and adaptive responses that influence cardiac repair. While estrogen has direct anti-inflammatory effects on blood vessels, women have a higher baseline immune setpoint, manifesting as more robust macrophage activation, greater antibody generation, with higher lymphocyte counts, lymphocyte to monocyte ratio, and higher adaptive immune response through Th1/Th17 polarization [117,135,136]. Due to stronger adaptive immune activation, women have an earlier reparative response but may also experience prolonged inflammation and consecutive fibrosis following MI [137]. Men, on the other hand, exhibit a more robust and earlier innate immune response, marked by increased neutrophil activation. Studies in murine models have shown that neutrophil counts post-MI were higher in male than female mice, which translates into a larger ischemic area, greater adverse remodeling, and a higher risk of heart failure with reduced ejection fraction [138,139]. Compared to young male mice, female mice have fewer neutrophils in the infarcted myocardium, but are more efficient at clearing debris and exhibit reduced pro-inflammatory gene expression, indicating better resolution of inflammation [138,139]. Estrogen appears to play a role in these differences also by promoting an anti-inflammatory and pro-resolving macrophage phenotype through ERα signaling, helping to limit excessive inflammation and fibrosis [140].

Overall men face a higher absolute risk and tend to experience cardiovascular disease earlier, but women’s relative risk rises significantly after menopause, and they often face worse outcomes once the disease develops.

## 6. Therapeutic Implications

As early as 1999, Ross et al. characterized atherosclerosis as a chronic inflammatory disease of the arterial wall, emphasizing that even after adjustment for traditional risk factors, such as elevated LDL cholesterol, a significant residual cardiovascular risk (RIR) remains [141]. This statement is derived from even earlier studies that established a clear association between systemic inflammation and increased cardiovascular risk. Elevated baseline plasma concentrations of CRP have been shown to independently predict incident atherothrombotic events, including myocardial infarction and ischemic stroke [142]. It is therefore unsurprising that, at that time, researchers started exploring anti-inflammatory therapy and hypothesized that medications already known to be effective in acute coronary events might also exert modulatory effects on vascular inflammation.

Presently, the importance of inflammation in cardiovascular pathology is further underscored by therapies targeting chronic inflammation and RIR in patients with established cardiovascular disease. The CANTOS trial (Canakinumab Anti-inflammatory Thrombosis Outcomes Study) on 10,061 patients with persistent inflammation following myocardial infarction demonstrated that inhibition of IL-1β reduced the risk of major adverse cardiovascular events by 15%, recurrent myocardial infarction by 24%, and cardiovascular mortality by 10% [143,144]. Beyond its impact on clinical outcomes, IL-1β blockade has been shown to modulate neutrophil biology, leading to attenuation of neutrophil-driven inflammation in clinical reports [145]. IL-1β is synthesized via NLRP3 inflammasome activation triggered by DAMPs, caspase-1 activation, and cleavage of pro-IL-1β into its biologically active form. Once released, IL-1β promotes endothelial activation, recruitment of neutrophils and monocytes into atherosclerotic plaques, and stimulates the production of IL-6 [146,147]. In CANTOS, this effect was reflected not only by reductions in inflammatory markers and RIR, but also by reductions in circulating neutrophil counts and an increased incidence of neutropenia in selected cases, findings consistent with suppression of myeloid activation as one of the mechanisms underlying the observed cardiovascular benefit [144]. By inhibiting pro-inflammatory signals that promote neutrophil recruitment and activation, IL-1β blockade indirectly limits neutrophil-monocyte cross-talk. Importantly, CANTOS provided the first robust evidence that reducing inflammation lowers cardiovascular risk independently of cholesterol levels [148]. Although promising, the results from CANTOS have not yet led to the clinical application of canakinumab in cardiovascular diseases, mainly due to safety concerns and the need to confirm its long-term efficacy. Targeting the IL-1β inflammatory pathway has been investigated in other studies. For instance, the VCU-ART 1, VCU-ART 2, and VCU-ART 3 trials conducted at Virginia Commonwealth University demonstrated that low-dose Anakinra, an interleukin-1 receptor antagonist, administered shortly after acute MI, significantly reduced circulating CRP levels. This reduction was further associated with a lower incidence of clinical events related to heart failure [149].

Between 2019 and 2021, several landmark trials repurposed colchicine, a well-established anti-inflammatory agent, demonstrating its efficacy in secondary prevention of cardiovascular events. Colchicine attaches to tubulin, the protein subunit of microtubules, thereby inhibiting microtubule polymerization. By disrupting microtubules, colchicine interferes with several important functions of innate immune cells, such as migration/chemotaxis, adhesion, degranulation, and NET formation in neutrophils, as well as impairing monocyte and macrophage motility and phagocytic activity [150,151]. The drug impairs neutrophil chemotaxis and endothelial adhesion, seemingly through effects on E- and L-selectin [152]. Colchicine inhibits the NLRP3 inflammasome in monocytes/macrophages by interfering with its assembly and function. Microtubule disruption prevents intracellular transport and prevents the close apposition of mitochondria and NLRP3, which is essential for activation. In addition to these effects, it interferes with pore formation mediated by purinergic P2X7 and P2X2 receptors, reducing pyroptosis and cytokine release [153]. Colchicine also reduces neutrophil-derived oxidative stress by interfering with early signaling events like tyrosine phosphorylation, preventing the assembly of NADPH oxidase and the production of superoxide [154]. These mechanistic insights offer the biological rationale behind the clinical benefits demonstrated in trials like COLCOT and LoDoCo2. The COLCOT trial (Colchicine Cardiovascular Outcomes Trial) revealed a notable decrease in cardiovascular events, such as heart attack, stroke, and urgent coronary revascularization, compared to placebo [155,156]. Similarly, the LoDoCo2 trial (Low-Dose Colchicine after Myocardial Infarction) showed that a daily dose of 0.5 mg of colchicine in patients with chronic atherosclerotic coronary disease significantly lowered the risk of ischemic events [156,157]. The results of LoDoCo2 and COLCOT diminished evidence gap, and based on these findings, the latest European guideline recommendations for chronic coronary syndromes suggest that the use of low-dose colchicine (0.5 mg/day) should be considered to lower the risk of myocardial infarction, stroke, and the need for revascularization (Class IIa recommendation) [158].

Based on previous findings, targeting inflammation has taken a step closer to becoming a strategy for cardiovascular disease, leading to extensive research for other inflammatory pathways. IL-6, an upstream mediator in the inflammatory cascade, is currently extensively studied as a potential therapeutic target in cardiovascular disease. Administration of a monoclonal antibody that blocks the interleukin-6 receptor, Tocilizumab, in the early hours after acute MI, as shown in the ASSAIL-MI trial, was found to decrease the extent of myocardial necrosis and lead to smaller injury areas according to cardiac MRI assessments, but raised safety concerns [159]. In RESCUE and RESCUE-2 trials, Ziltivekimab, an investigational monoclonal antibody targeting IL-6, significantly reduced hsCRP, fibrinogen, and serum amyloid A, compared to placebo [160,161]. Ziltivekimab administration did not produce a notable rise in adverse events like thrombocytopenia, neutropenia, or infections, the main concerns in other IL-6 inhibitors such as tocilizumab [159,161]. Inhibiting IL-6 may reduce endothelial dysfunction, reduce the recruitment of monocytes and neutrophils into atherosclerotic plaques, and minimize the risk of plaque destabilization and thrombosis [162]. Ziltivekimab is at the moment an investigational therapy for cardiovascular purposes, and its study is now expanded to multiple scenarios, such as chronic atherosclerotic disease (ZEUS) [163], acute myocardial infarction (ARTEMIS) [164], and chronic heart failure (HERMES) [165].

Other anti-inflammatory strategies that have gained significant attention in recent years focus on targeting the NLRP3 inflammasome, a central regulator of innate immunity. Inhibitors of NLRP3 are considered among the most promising next-generation therapeutic approaches, offering the potential to suppress detrimental inflammation while preserving host defense. By acting upstream, NLRP3 inflammasome inhibitors may provide suppression of harmful inflammation. Among them, MCC950 stands out as a highly selective compound that inhibits ATP hydrolysis and prevents the assembly of the NLRP3 complex, thereby effectively blocking inflammasome activation. In preclinical studies on conditions like myocardial infarction, heart failure, and atherosclerosis, MCC950 showed impressive results by reducing IL-1β production, decreasing neutrophil infiltration, and limiting harmful cardiac remodeling. This made it one of the most promising early agents targeting NLRP3 [166,167]. However, its clinical development was stopped due to concerns about liver toxicity [168]. MCC950 represented a proof-of-concept compound, proving that targeting NLRP3 for treating cardiovascular disease and other inflammatory conditions is possible, and its development influenced research for creating safer, next-generation inhibitors with better pharmacological profiles. Therefore, in murine models of myocardial ischemia–reperfusion injury, administration of an orally available NLRP3 inhibitor, OLT1177 (dapansutrile), within 60 min of reperfusion significantly limited necrosis and consecutively prevented the development of secondary myocardial fibrosis and LV systolic dysfunction in a dose-dependent manner, through selective inhibition of the NLRP3 inflammasome [169]. New NLRP3 inhibitors like inzomelid (DFV890, Novartis) and agents derived from somaliximab are currently being studied in early-phase clinical trials. Preliminary results indicate they have promising anti-inflammatory effects and good tolerability profiles [170,171].

Sodium–glucose cotransporter 2 (SGLT2) inhibitors, originally developed to lower blood sugar, have become a game changer in modern medicine. This drug significantly reduces mortality and major cardiovascular events, not just in people with type 2 diabetes but also in those without diabetes [172,173,174]. Various studies and meta-analyses have shown that SGLT2 therapy can reduce levels of IL-6, TNF-α, MCP-1, and hsCRP, as well as NLRP3 downstream activation factors, although the findings are still heterogeneous [175,176,177]. In addition, SGLT2 inhibitors have been shown to attenuate NET formation in models of cardiac injury [178], reduce the infiltration of monocytes and macrophages into atherosclerotic lesions [179] and reprogram macrophage metabolism toward an anti-inflammatory phenotype [180]. Small preclinical studies suggest that empagliflozin helps restore neutrophil metabolic activity and improves their functional capacity by lowering plasma levels of 1,5-anhydroglucitol. Moreover, it down-regulates heart HMGB1 expression and attenuates NET formation and cardiac fibrosis [178].

Targeting NET formation via PAD4 inhibition or enhancing NET degradation by DNase therapy has also been suggested as a new anti-inflammatory treatment approach for vascular inflammation and thrombosis. DNase therapy in preclinical models reduces thrombosis and infarct size [181,182,183] diminished oxidative stress and improved endothelial function [184,185]. Recent clinical studies have highlighted the potential of DNase therapy in overcoming thrombolytic therapy resistance [184,186]. Inhibiting PAD4 directly diminished NET formation, lowered neutrophil infiltration in the myocardium, and reduced the production of IL-1, IL-6, and TNF-α [187,188].

In addition to targeting inflammation, recent therapeutic strategies have increasingly focused on cell-specific modulation of myeloid cell recruitment and function after MI by targeting CCR2 involved in the recruitment of pro-inflammatory Ly6C^high^ monocytes, or by promoting MerTK signaling, a receptor tyrosine kinase essential for efferocytosis and inflammation resolution. For instance, inhibition of CCR2 with propagermanium leads to altered distribution of macrophage subsets and favorable tissue remodeling after MI [189] and targeted deletion of CCR2 attenuated left ventricular remodeling after MI in mice [190]. Moreover, efferocytosis is a prerequisite for inflammation resolution and tissue repair [16] and MerTK activation can be achieved indirectly via endogenous ligands such as Gas6 and Protein S, which enhance apoptotic cell clearance and trigger pro-resolving macrophage reprogramming [191]. These data highlight how selective modulation of monocyte recruitment and macrophage reprogramming may complement traditional therapies by restoring immune homeostasis after ischemic injury.

Despite progress in standard treatments, residual risk persists in atherosclerotic disease and heart failure. Anti-inflammatory treatments show promise in addressing this issue, but their use must carefully balance protective and harmful inflammation, particularly in acute settings where it occurs in unison. Recent research has demonstrated that neutrophil–macrophage crosstalk plays a central role in modulating vascular inflammation and atherothrombosis, highlighting this cellular interaction as a key target for future therapeutic interventions. Emerging strategies will likely focus on timing, patient selection, and tailored therapies with targeted small molecules, and nanoparticle-based delivery systems to incorporate precise anti-inflammatory therapy into personalized cardiovascular care.

## 7. Gaps in Knowledge—Future Directions

Despite significant advances in understanding how neutrophils modulate macrophage phenotypes post-injury, several important gaps remain, as follows:(i).Most studies focus on static images of neutrophil–macrophage interactions, often in vitro or at single time points in vivo. The exact timing and sequence of neutrophil-derived signals (cytokines, chemokines, granule proteins, NETs, and EVs) that dictate macrophage polarization in different phases of cardiac injury remain poorly defined. Moreover, both neutrophils and macrophages are highly heterogeneous, with subpopulations exhibiting distinct functional profiles. How specific neutrophil and macrophage distinct subsets influence each other post-myocardial infarction is largely unexplored.(ii).While DAMPs and EVs have been shown to activate macrophages via TLR4 or other pathways, the downstream signaling networks, cross-talk with other immune cells, and contribution to maladaptive remodeling are not fully understanded. Additionally, the cargo composition of EVs in different activation states and its functional consequences require further investigation.(iii).Despite recognition of neutrophil-derived mediators as modulators of macrophage phenotype, strategies to selectively manipulate these signals without compromising host defense are still in their infancy. Optimizing timing, delivery, and specificity of interventions targeting neutrophil–macrophage communication is a major unmet need.(iv).When considering early therapeutic interventions to modulate the neutrophil–macrophage cross-talk, the cardiac resident macrophages may represent a primary target that was not investigated until now.

Addressing these gaps will be critical to progress in both understanding neutrophil-mediated regulation of inflammation and repair post-MI and to develop targeted therapies for cardiovascular injury.

## 8. Conclusions

Neutrophils and macrophages are highly heterogeneous, and their shifting balance throughout the progression from inflammation to repair is critical in modulating the extent of collateral damage or healing in the infarcted heart. Shortly after myocardial infarction, neutrophils and macrophages infiltrate the ischemic tissue, and a dynamic cross-talk between them critically shapes the trajectory of myocardial infarction healing, from the initial inflammatory burst to resolution and scar formation. While neutrophils rapidly infiltrate and influence monocyte-derived macrophage polarization, evidence also points to an initiating role of resident macrophages in orchestrating neutrophil recruitment. This bidirectional communication ensures efficient debris clearance and sets the stage for reparative remodeling; however, excessive or prolonged signaling may contribute to adverse remodeling and heart failure. Deciphering the temporal and functional diversity of neutrophil–macrophage interactions can not only advance our mechanistic understanding of cardiac inflammation but also unveil novel therapeutic opportunities by targeting specific phases or cellular subtypes, which could improve post-MI outcome.

## Figures and Tables

**Figure 1 ijms-26-10575-f001:**
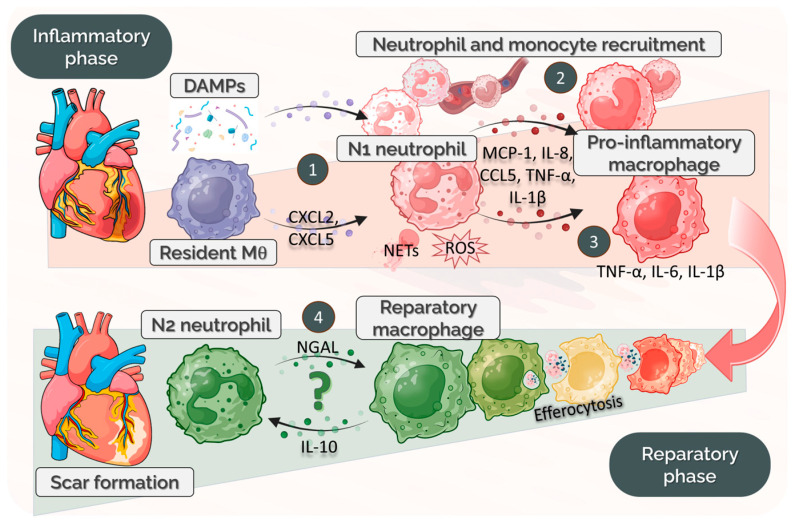
Schematic representation of the temporal dynamics of neutrophil–macrophage communication following myocardial infarction (MI). Following MI, DAMPs released from necrotic tissue activate resident CCR2^+^ macrophages, which secrete inflammatory cytokines and chemokines that recruit circulating neutrophils to the infarcted area, marking the first level of intercellular communication between macrophages and neutrophils ①. Infiltrating neutrophils produce chemokines (MCP-1, CCL5) that attract CCR2^+^Ly6C^+^ monocytes ②, which subsequently differentiate into pro-inflammatory macrophages as a result of neutrophil secreted pro-inflammatory stimuli, marking a third level of communication ③. During the transition from the inflammatory to the proliferative phase (days 4–7 post-MI), macrophages adopt anti-inflammatory and pro-reparative profiles, contributing to tissue remodeling, angiogenesis, and fibrosis. Neutrophil-derived mediators, including NGAL and NET-associated DNA, modulate macrophage efferocytosis and polarization, marking the fourth level of communication ④. Emerging evidence suggests the existence of reparative neutrophil subtypes (N2) that may interact bidirectionally with reparatory macrophages and vice versa to coordinate late-phase healing and scar formation, although further research is required.

## Data Availability

No new data were created or analyzed in this study. Data sharing is not applicable to this article.

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
