# Peer review of "Cross-Talk Between Neutrophils and Macrophages Post-Myocardial Infarction: From Inflammatory Drivers to Therapeutic Targets"

_ijms, 2025, doi:10.3390/ijms262110575_

Round 1

Reviewer 1 Report

Comments and Suggestions for Authors

Thank you for very nice written and comprehensive review. I have only one suggestion. Please expand the figure legend. In the text it was mentioned three levels of neutrophil-macrophage communication, please, indicate them on the figure and describe in more details in the figure legend that the readers should not go back to the text where it is presented.

Author Response

We thank the Reviewer for their positive feedback on our work and for the suggestion regarding the figure legend.

Comment

Thank you for very nice written and comprehensive review. I have only one suggestion. Please expand the figure legend. In the text it was mentioned three levels of neutrophil-macrophage communication, please, indicate them on the figure and describe in more details in the figure legend that the readers should not go back to the text where it is presented.

Response

According to the reviewer's suggestion, we have expanded the legend of Figure 1 to provide a more detailed and self-explanatory description. Specifically, we have indicated the four levels of neutrophil–macrophage communication directly within the figure and described each level in detail in the revised legend.

Reviewer 2 Report

Comments and Suggestions for Authors

The review by Letitia Ciortan at al. provides a well-structured overview of neutrophil–macrophage interactions following myocardial infarction, and it succeeds in summarizing key inflammatory mechanisms underlying post-ischemic injury and repair. However, several important aspects could be expanded or refined to enhance the depth, clarity, and impact of the manuscript.

  • The review focuses mainly on cytokine- and receptor-mediated communication but omits discussion of metabolic crosstalk between neutrophils and macrophages, including how metabolites such as lactate, succinate, or arginine derivatives influence macrophage polarization and resolution. This aspect of immunometabolism has gained major attention in recent years and would substantially enrich the mechanistic depth of the review. In addition, there is no discussion of sex- or age-related differences in immune responses after myocardial infarction, although these are increasingly recognized as key determinants of repair efficiency.
  • Likewise, while the review mentions single-cell transcriptomic studies, it does not discuss how emerging spatial transcriptomics technologies have refined our understanding of myeloid cell interactions in situ.
  • The “Therapeutic Implications” section could be expanded to include recent developments, such as cell-targeted delivery systems, CCR2 inhibition, MerTK agonists.
  • Summarizing complex pathways or molecular mediators in tables or schematic diagrams would aid comprehension.
  • The review would benefit from the inclusion of the following study: Ferraro B, Leoni G, Hinkel R, Ormanns S, Paulin N, Ortega-Gomez A, Viola JR, de Jong R, Bongiovanni D, Bozoglu T, Maas SL, D'Amico M, Kessler T, Zeller T, Hristov M, Reutelingsperger C, Sager HB, Döring Y, Nahrendorf M, Kupatt C, Soehnlein O. Pro-Angiogenic Macrophage Phenotype to Promote Myocardial Repair. J Am Coll Cardiol. 2019 Jun 18;73(23):2990–3002. doi: 10.1016/j.jacc.2019.03.503. PMID: 31196457. This study shows that neutrophil-derived annexin A1 promotes cardiac repair by driving macrophages toward a pro-angiogenic phenotype that releases VEGF-A, thereby enhancing neovascularization and improving heart function.

Author Response

We thank the Reviewer for their thorough and constructive comments, which have significantly contributed to improve our manuscript. All suggestions have been carefully addressed, and the corresponding modifications are highlighted in blue in the revised version.

Specifically:

Comment 1

The review focuses mainly on cytokine- and receptor-mediated communication but omits discussion of metabolic crosstalk between neutrophils and macrophages, including how metabolites such as lactate, succinate, or arginine derivatives influence macrophage polarization and resolution. This aspect of immunometabolism has gained major attention in recent years and would substantially enrich the mechanistic depth of the review.

Response 1

Thank you for this comment; this is really an important aspect we have missed in the first variant. We have now included a new subsection “4.5. Metabolic Crosstalk Between Neutrophils and Macrophages” discussing metabolic crosstalk between neutrophils and macrophages, emphasizing how key metabolites (e.g., lactate, succinate, and arginine derivatives) regulate macrophage polarization and resolution dynamics, at pages 14-15, lines 603-668.

Comment 2

In addition, there is no discussion of sex- or age-related differences in immune responses after myocardial infarction, although these are increasingly recognized as key determinants of repair efficiency.

Response 2

According to the reviewer suggestion, a new section “5.3. Age- and sex-driven immune dysregulation in cardiovascular injury” has been added and discussed on sex- and age-related differences in immune responses after myocardial infarction, highlighting how these factors influence inflammation and repair (pages 17-18, lines 768-827).

Comment 3

Likewise, while the review mentions single-cell transcriptomic studies, it does not discuss how emerging spatial transcriptomics technologies have refined our understanding of myeloid cell interactions in situ.

Response 3

The reviewer suggestion is pertinent. We have expanded the discussion of single-cell transcriptomic studies with existent data emerging from spatial transcriptomics technologies, at pages 8-9, lines 361-384.

Comment 4

The “Therapeutic Implications” section could be expanded to include recent developments, such as cell-targeted delivery systems, CCR2 inhibition, MerTK agonists.

Response 4

The “Therapeutic Implications” section has been revised to incorporate recent strategies, including cell-targeted delivery systems, CCR2 inhibition, and MerTK agonists. (page 21, lines 957-967)

Comment 5

Summarizing complex pathways or molecular mediators in tables or schematic diagrams would aid comprehension.

Response 5

To facilitate comprehension, we have added a new table (pages 10-11) summarizing the key molecular mediators that impact macrophage/neutrophil behavior and polarization.

Comment 6

The review would benefit from the inclusion of the following study: Ferraro B, Leoni G, Hinkel R, Ormanns S, Paulin N, Ortega-Gomez A, Viola JR, de Jong R, Bongiovanni D, Bozoglu T, Maas SL, D'Amico M, Kessler T, Zeller T, Hristov M, Reutelingsperger C, Sager HB, Döring Y, Nahrendorf M, Kupatt C, Soehnlein O. Pro-Angiogenic Macrophage Phenotype to Promote Myocardial Repair. J Am Coll Cardiol. 2019 Jun 18;73(23):2990–3002. doi: 10.1016/j.jacc.2019.03.503. PMID: 31196457. This study shows that neutrophil-derived annexin A1 promotes cardiac repair by driving macrophages toward a pro-angiogenic phenotype that releases VEGF-A, thereby enhancing neovascularization and improving heart function.

Response 6

As suggested by the reviewer, we have added Annexin A1 as a mediator from neutrophil EVs that impact macrophages, thus included and discussed the study by Ferraro et al., JACC 2019, as suggested, in the “Extracellular vesicles” section, page 13, lines 571-578.

Reviewer 3 Report

Comments and Suggestions for Authors

General Comments

This review by Ciortan et al., entitled “Cross-Talk Between Neutrophils and Macrophages Post-Myocardial Infarction: From Inflammatory Drivers to Therapeutic Targets” focused on the power of immuno-inflammation to influence the pathogenetic chain leading to myocardial infarction and its complications. This is an emerging topic of pathophysiology and  an impactful platform on which new therapeutic strategies are being planned. However, even to correctly address the design of new therapeutic strategies, the focus can not be restricted only to innate immunity, where neutrophils and macrophages are major players. In fact, the final phenotype determining multi-organ damage is the derangement of interplay between innate and adaptive immunity (see for review Buonacera et al., Int J Mol Sci 2002), also involving lymphocytes, which work as major actors of adaptive immunity. Therefore, this review, while shedding light on the interaction between neutrophils and macrophages (innate immunity), should also consider their interactions with lymphocytes, so enlarging the focus on the adaptive immunity. In this direction, the importance of the role of neutrophil-to-lymphocyte ratio (NLR) as a mirror of the derangement in the interplay between innate and adaptive immunity, that is a major driver to tissue damage, shoud not be left out.

Specific Comments

Bearing in mind that very often in clinical practice both diagnostic and theraputic strategies would result successfully if standing on a well balanced pathophysiologic platform, Authors are kindly requested to enlarge the focus of this review on both phases of immunity, also taking into account previous evidence supporting this strategy to identify carotid atherosclerosis, even in older individuals (Corriere T et al.,Nutr Metab cardiovasc Dis 2018), as well as to independently forecast the risk of MACE during hospitalization for patients with STEMI (Xin J et al., BMJ Open 2025).

Author Response

We thank the Reviewer for their thorough and constructive comments, which have significantly contributed to improve our manuscript. The suggestions have been carefully addressed, and the corresponding modifications are highlighted in blue in the revised version.

Comment

However, even to correctly address the design of new therapeutic strategies, the focus can not be restricted only to innate immunity, where neutrophils and macrophages are major players. In fact, the final phenotype determining multi-organ damage is the derangement of interplay between innate and adaptive immunity (see for review Buonacera et al., Int J Mol Sci 2002), also involving lymphocytes, which work as major actors of adaptive immunity. Therefore, this review, while shedding light on the interaction between neutrophils and macrophages (innate immunity), should also consider their interactions with lymphocytes, so enlarging the focus on the adaptive immunity. In this direction, the importance of the role of neutrophil-to-lymphocyte ratio (NLR) as a mirror of the derangement in the interplay between innate and adaptive immunity, that is a major driver to tissue damage, shoud not be left out.

Bearing in mind that very often in clinical practice both diagnostic and theraputic strategies would result successfully if standing on a well balanced pathophysiologic platform, Authors are kindly requested to enlarge the focus of this review on both phases of immunity, also taking into account previous evidence supporting this strategy to identify carotid atherosclerosis, even in older individuals (Corriere T et al.,Nutr Metab cardiovasc Dis 2018), as well as to independently forecast the risk of MACE during hospitalization for patients with STEMI (Xin J et al., BMJ Open 2025).

Response

We understand the importance of neutrophil- and macrophage-lymphocyte interactions in orchestrating the overall inflammatory and reparative response, and in line with the Reviewer’s suggestion, a short passage was added in section 1, lines 78-81 and a discussion addressing the connection between innate and adaptive immunity, including the clinical relevance of the neutrophil-to-lymphocyte ratio (NLR), has been added in the new section 5.3 on pages 17-18, lines 768-827 of the revised manuscript. The section also includes the data from the suggested paper of Buonacera et al., Int J Mol Sci 2002.

In addition, we included references to the studies Corriere et al., Nutr Metab Cardiovasc Dis 2018, and Xin et al., BMJ Open 2025 (page 18, lines 798 and 802) — to highlight the importance of NLR as a strong predictor of the presence and number of carotid atherosclerotic plaques in older adults, as well as the feature that it can be used independently to forecast the risk of major adverse cardiovascular events (MACE) during hospitalization.

Round 2

Reviewer 3 Report

Comments and Suggestions for Authors

This review now sounds very well and looks based on a holisitic vision of  of the impact of immunity on post-ischemic residual risk. I appreciated it.